# Gender Differences in microRNA Expressions as Related to Long-Term Graft Function in Kidney Transplant Patients

**DOI:** 10.3390/ijms232112832

**Published:** 2022-10-24

**Authors:** Sien-Yu Ko, Shang-Feng Tsai, Chia-Tien Hsu, Shih-Ting Huang, Ya-Wen Chuang, Tung-Min Yu, Ming-Ju Wu, Cheng-Hsu Chen

**Affiliations:** 1School of Medicine, Chung Shan Medical University, Taichung 40201, Taiwan; 2Division of Nephrology, Department of Internal Medicine, Taichung Veterans General Hospital, Taichung 407219, Taiwan; 3Department of Post-Baccalaureate Medicine, College of Medicine, National Chung Hsing University, Taichung 40227, Taiwan; 4Department of Life Science, Tunghai University, Taichung 407224, Taiwan; 5School of Medicine, China Medical University, Taichung 404333, Taiwan

**Keywords:** kidney transplant, microRNA, immunosuppression

## Abstract

In recent studies, much has been discussed about biomarkers used in the evaluation of the transplanted graft function. However, there remains a lack of research regarding the long-term effects of microRNAs (miRNAs) on the different genders for kidney transplant (KTx) patients. In this study, we aim to assess the functions of miRNAs on long term outcomes of KTx patients by extracting differently expressed miRNAs between patients of normal graft function and graft dysfunction, while further analyzing their impact on the different genders. We analyzed the data of 40 patients who had received KTx for a period of more than ten years and included data regarding renal function, immuno-related markers and plasma miRNAs. Data were classified by gender for further studies. Twelve out of 17 females and 8 out of 23 males had undergone graft dysfunction. Renal function analysis showed significantly worse outcomes in the female patients. There were five differently expressed miRNAs between the female control group and female dysfunction group: miR-128-3p, miR-21-5p, miR-150-5p, miR-92a-3p and miR-15a-5p, and five between the male control group and male dysfunction group: miR-23a-3p, miR-126-3p, miR-142-3p, miR-223-3p and miR-26a-5p. Gender differences exist in incidences of kidney graft dysfunction, with male patients displaying better preservation in graft functions. Overall, these differently expressed miRNAs either enhance or suppress host immune responses. They can be predictive markers for graft survival and can also be important factors that lead to worse long term kidney graft function in females when compared to males.

## 1. Introduction

End-stage kidney disease (ESKD) indicates irreversible kidney function impairment, the patients of which permanently rely on hemodialysis, peritoneal dialysis or kidney transplantation (KTx) in order to sustain life [1,2]. Aiming at both a better quality of life and the reduction of mortality risk, KTx has long been an alternative therapy for ESKD patients.

Studies have shown that differences in hormones and genes are considered to have an impact on diseases and health status [3,4]. Recently, expressions of microRNA (miRNA) have been particularly highlighted due to its various effects on downstream signal pathways [5], including proinflammatory cytokines and anti-inflammatory cytokines [6]. MiRNAs are groups of non-coding RNAs possessing essential cellular functions such as cell proliferation, cell differentiation, organ development and immuno-regulation [7]. Their various functions make them ideal non-invasive markers during certain evaluations or follow ups [8,9]. Renal graft biopsies are the current gold standard for the examination of entities involving renal graft dysfunction. However, biopsies are both expensive and cause tissue damage, not to mention the associated risks related to any complications. The characteristics of being convenient, minimally-invasive and functionally variable make miRNAs ideal biomarkers for the prediction of long-term graft function.

MiRNAs play a crucial role in explaining both different disease progressions and prognosis between the two genders due to their uneven distribution within the sex chromosomes and autosomes [10,11,12,13]. The expression of miRNAs has been particularly highlighted due to their various effects on downstream signal pathways, including immuno-related cytokines. They therefore play a crucial role in explaining the different results of long-term graft function between the two genders due to their uneven distribution not only on sex chromosomes but also on autosomes.

Much has been discussed about the various biomarkers being used for the evaluation of transplanted graft function, but there has been a lack of studies surrounding the role of miRNAs in the different genders of KTx patients, particularly regarding their impact on maintaining long-term graft function. In this study, we aim to assess the functions of miRNAs on long term outcomes of KTx patients by extracting differently expressed miRNAs between patients of normal graft function and graft dysfunction, while also further analyzing their impact on the different genders.

## 2. Results

### 2.1. Patient Characteristics

A demographic analysis of the 40 patients with KTx over a period of more than ten years is shown in Table 1 and Figure 1. Amongst all 40 patients, 12 out of 17 females and eight out of 23 males had previously undergone graft dysfunction. The graft dysfunction rate of the female KTx patients (71%) was shown to be significantly higher than that of male patients (35%), with a *p*-value < 0.05. The mean age of participants is oldest in the female control group and youngest in the male dysfunction group. Age when having received transplant shows similar results, with the oldest age being seen in the female control group and the youngest age seen in the male dysfunction group. Graft survival time length was calculated by subtracting the patient’s age when receiving transplant from their current age. This was shown to be longest in the female control group and shortest in the male dysfunction group.

### 2.2. Renal Function

Data regarding renal function are also analyzed in Table 1. BUN, creatinine and eGFR show significant differences between the groups. BUN level was high in the female dysfunction (25.9 ± 10.1) and male dysfunction groups (20.6 ± 3.3), and low in the female control (13.6 ± 2.7) and male control groups (20.6 ± 3.3). Creatinine levels show a similar pattern. The estimated glomerular filtration rate (eGFR) was high in the male control (75.2 ± 7.2) and female control groups (72.0 ± 8.9) and low in the male dysfunction (48.6 ± 7.2) and female dysfunction groups (40.8 ± 9.5). The urine protein/creatinine ratio was high in both dysfunction groups, particularly in females.

### 2.3. Immune Factors

Complete blood counts and other immune related factors are investigated and shown in Table 1. However, none showed any significant difference. Drug regimens used in immunosuppression therapy varied between patients, but serum concentrations of FK-506 were similar. The daily dosage of adagraf/prograf was significantly different between the groups of patients, with a *p*-value of 0.03. FK-506 concentrations didn’t show any significant difference between the groups, with a *p*-value of 0.57.

### 2.4. Differently Expressed miRNAs

Differently expressed miRNAs in the female and male dysfunction groups are summarized in Table 2 and Table 3. Mean Ct value of differently expressed miRNAs in female and male patients are shown in Figure 2 and Figure 3. The standard selection criteria to identify differently expressed genes were delta Cq ≥ |0.585| and an adjusted *p* < 0.05. There were 5 miRNAs expressing with significant difference between the female control group and female dysfunction group: miR-128-3p, miR-21-5p, miR-150-5p, miR-92a-3p and miR-15a-5p. In the female dysfunction group, miR-128-3p was up-regulated, while others were down-regulated, when compared to the female control group. There were 5 expressing with significant difference between the male control group and male dysfunction group: miR-23a-3p, miR-126-3p, miR-142-3p, miR-223-3p and miR-26a-5p. All of them were down-regulated in the male dysfunction group when compared to the male control group. After miRNA analysis of all 40 patients, these 10 miRNAs were highlighted due to significantly different expressions between the control group and graft dysfunction group in either gender.

## 3. Discussion

Various disease manifestations in both male and female patients need to be better understood in order to evaluate disease progression and to determine the ideal medical intervention. Studies have shown that differences in hormones and genes are considered to have an impact on both disease and health status [3]. Recently, expressions of miRNAs have been particularly highlighted due to their various effects on downstream signal pathways [5], including immuno-related cytokines [6]. To explain the reason for generally worse long-term outcomes in female KTx patients, we investigated the pathology of kidney graft injuries and collected data regarding immune-related serum miRNA to study the possible correlation.

The main cause of graft dysfunction lies in chronic cellular and humoral immune responses, which will likely lead to kidney graft injuries. T cells activate when regarding a transplanted kidney as being foreign and stimulate the downstream immune pathway, which can directly damage graft tissues [14]. Dendritic cells play the role of antigen presenting cells and act as the bridge between innate and adaptive immune responses. Natural killer cells increase the kidney infiltration of immune cells and cause tissue damage [15]. In other words, inflammation places KTx patients at risk of graft dysfunction. To explain why female patients in our study experienced a generally worse long-term outcome, multiple inflammation inducing factors should be studied further. Here, we focus on expressions of immune-related serum miRNA in long term graft dysfunction and investigate their effects on renal function of different genders. Differences in miRNA expressions, inflammatory pathways and cytokine production should be considered.

MiRNAs are groups of noncoding RNAs with essential cellular functions such as cell proliferation, cell differentiation, organ development and immuno-regulation. The expression of miRNAs has been particularly highlighted due to their various effects on downstream signal pathways, including immuno-related cytokines. They therefore play a crucial role in explaining the different results of long-term graft function between the two genders due to their uneven distribution not only on sex chromosomes but also on autosomes [12,13,16].

This brings out another finding taken from our study: in dysfunction groups, there are some differently expressed miRNAs, on either autosomes or sex chromosomes, whose functions are to regulate inflammatory related cytokines. Through statistical analysis, we identified miRNAs that displayed significantly different expressions between the control and dysfunction groups of the same gender. Five (5) miRNAs have been noted in the female groups and another five in the male groups. We summarize the functions of those highlighted miRNAs in Table 4.

Between the female control group and female dysfunction group, five miRNAs are expressed with a significant difference: miR-128-3p, miR-21-5p, miR-150-5p, miR-92a-3p and miR-15a-5p. MiR-128-3p is up-regulated, while the others are down-regulated in the female dysfunction group. MiR-92a-3p is located on the X chromosome while others are found on the autosomes.

Between the male control group and male dysfunction group, there are also 5 miRNAs expressing with a significant difference: miR-23a-3p, miR-126-3p, miR-142-3p, miR-223-3p and miR-26a-5p. All of them are down-regulated in the male dysfunction group compared to the male control group. MiR-223-3p is located on the X chromosome, while others are found on the autosomes.

As the results reveal, those microRNAs are significant in either the female or male groups. None of them are significant in both groups, indicating that there are existing gender differences in the microRNA expression of graft dysfunction patients.

Though the roles of the differently expressed microRNAs are not yet fully understood, some theories have been introduced to better clarify their physiological functions, particularly regarding immunity. 

MiR-128-3p controls innate immunity, positive regulation of macrophage differentiation and the NF-κB signaling pathway. The down-regulation of miR-128-3p suppressed inflammation and the NF-κB pathway [17]. Studies have revealed that an increased expression of miR-128-3p is accompanied by increased levels of IL-6 and IL-17 in T cells [17], while a decreased expression of miR-128-3p can suppress p-IkBα and CD69, and CD25 expression in T cells. Though one of the limitations of our study is the absence of data regarding pro-inflammatory cytokines such as IL-6, several studies have lab results confirming this. The study of Xia et al. supports this point. It was mentioned that miR-128-3p played a similar role in patients diagnosed with rheumatoid arthritis (RA), whose immune response was suppressed and RA symptoms were mediated when expression of miR-128-3p decreased [17]. It was found that significant increase of miR-128-3p accompanied the activation of T cells, pro-inflammatory cytokines and the NF-κB signaling pathway. The downregulation of miR-128-3p, on the other hand, mediated progression of RA by suppressing inflammation and the NF-κB signaling pathway. In addition, a study of Xiaokelaiti et al. proved that the level of pro-inflammatory cytokine IL-6 decreased while anti-inflammatory cytokine IL-10 elevated after the transfection of miR-128-3p [18]. They furthermore found corresponding results when IL-6 increased and IL-10 decreased in miR-128-3p silencing cells. This could explain our result showing the female dysfunction group with upregulated miR-128-3p, which can exacerbate any inflammation response and lead to worse long term graft function. 

MiR-150-5p plays a crucial role in the immune regulation of B and T lymphocytes, controlling their apoptosis, activation and proliferation. MiR-150 also displays selective expression, particularly in immune organs such as lymph nodes, B and T lymphocytes and spleen [19]. One pathway of miR-150 is to inhibit the expression of Myb10 in order to further limit B lymphocyte proliferation [20]. That is to say, with a lack of miR-150, the immune response of CD8+ T cells would be more enhanced during activation [19]. Another pathway of miR-150 is to cooperate with miR-99a and suppress the mammalian target of rapamycin (mTOR) expression, whose function is to inhibit Treg differentiation [21]. Therefore, a downregulated miR-150 indicates both a higher level of mTOR expression and more inhibited Treg differentiation. Treg cells are responsible for suppressing immune responses by inhibiting cytokine production and T cell proliferation. In transplant patients, immune suppression status is especially important for preventing graft rejections. This could explain why female patients with a worse long-term graft function have shown a significantly lower expression of miR-150 in our study [9].

MiR-223-3p has been known for its regulative function on the differentiation of immune cells and the secretion of inflammatory cytokines. Proliferation and differentiation of granulocytes, macrophages and dendritic cells are significantly related to miR-223-3p expression [22]. Research has pointed out that miR-223 negatively regulates granulocyte differentiation and activation. Johnnidis et al. designed an experiment involving miR-223 mutant (loss of function) mice [23]. Granulocytes without miR-223 are found to be over-mature and overreact to immune responses. Consequently, miR-223 mutant mice tend to develop inflammatory pathology and tissue damage in organs [23]. In macrophages, a high expression of miR-223 suppresses TNF receptor-associated factor 6 (TRAF6), inhibits the NF-κB signaling pathway and mediates tissue injury caused by inflammation. The down regulation of miR-223, on the other hand, is known to induce an inflammatory response in particular tissues. In dendritic cells, miR-223-3p attenuates their ability for antigen uptake and antigen presenting, meanwhile stimulating Treg cell differentiation [24]. miR-223-3p also regulates inflammatory related cytokines such as TNF-α, IL-6, IL-8, and IL-1β, with IL-6 being a key factor in allograft injury [25,26,27]. Miller et al. have pointed out in their research that IL-6 promotes T cell activity and causes the impairment of Treg cells in mouse skin transplant models [27]. Similarly, in transplant patients, it has been revealed that higher levels of IL-6 gene expression and protein amounts are related to worse graft functions. That is, elevated levels of IL-6 in serum and kidney tissue are found in patients experiencing kidney graft rejection. Graft rejection is usually related to inflammatory responses, which has been observed to correlate with the degree of increased IL-6 levels [27]. There are several pathways related to the regulation of IL-6 production for mediating immune responses. IL-6 secretion in mast cells can be reduced when the IGF1R/PI3K signaling pathway is suppressed by miR-223-3p [26]. To summarize, it can be explained that the down-regulation of miR-223-3p resulted in graft dysfunction of our male patients due to its regulative function on the differentiation of immune cells and secretion of inflammatory cytokines.

The role of miR-26a as a critical regulator of renal physiology and pathology has been highlighted recently. Its protective functions regarding the kidney include maintaining the biological function of podocytes, mesangial cells and T cells in the kidney. MiR-26a has also been known to inhibit pro-inflammatory cytokine production by suppressing the NF-κB pathway [28]. In the study of Chen et al., it was proven that the suppressing miR-26a-5p promoted kidney inflammation and renal cell injury [29]. The induction of miR-26a-5p resulted in the inhibition of IL-6 and thus alleviated renal inflammation. Another function of miR-26a is to reduce the risk of renal ischemia-reperfusion injury (IRI), which is also a leading cause of graft dysfunction after transplant [30]. IRI can cause renal tubular injury due to the infiltration of immune cells, such as neutrophils and macrophages. Here, the overexpression of miR-26a acts as a protective factor to induce Treg proliferation and attenuate renal IRI. In the study of Liang et al., miR-26a also stimulates Tregs/CD4+ T cells ratio and reduces IL-6 secretion [31]. Thus, the down-regulation of miR-26a seen in our male dysfunction patients could mean a diminished protection of kidney grafts.

The small population of our patients limited the level of evidence we could achieve. Furthermore, some functions of specific miRNAs have not yet been well clarified. Analyzing the levels of pro-inflammatory cytokines in different groups of patients would be our goal in our next study to prove our points. More data regarding long-term KTx patients are still required in order to better analyze the relationship between differently expressed microRNAs and kidney graft functions.

## 4. Patients and Methods

### 4.1. Data and Participants

To better understand long-term outcomes of patients who had received KTx, we analyzed the follow up data of 40 patients who had received KTx for more than ten years, with either deceased or living donors. The enrolled KTx recipients had been routinely followed up at our department for information regarding graft function, immune status and graft survival. We included data on current age, age receiving KTx, duration of graft survival, renal graft function (BUN, Creatinine and eGFR), UPCR, WBC, Hb, neutrophil, lymphocyte, N/L ratio, platelet, P/L ratio, humoral immunity and cellular immunity. Humoral immunity was presented as a complement activation (C3, C4 and CH50) and principal B-cell function, as indicated by levels of antibody immunoglobulins (IgG, IgA and IgM). Cellular immunity was presented through lymphocyte subset analysis, including CD3, CD19, CD4, CD8, NK, LEU-3A (CD4), LEU-2B (CD8) and HLA-DR. Drug regimens involving immunosuppression therapy after KTx were recorded as daily dosages of Adagraf/prograf (mg/day). Serum concentrations of FK-506 were investigated and analyzed. Graft dysfunction was defined as an eGFR lower than 60 mL/min. The study was conducted according to the guidelines of the Declaration of Helsinki and approved by the Ethics Committee of the Taichung Veterans General Hospital (IRB No.: CE18192A).

### 4.2. Serological Assessment

There were 89 miRNAs used for analysis and they were selected using text mining. The functions of these miRNAs include immune, inflammation and cell differentiation. Blood was drawn from 40 KTx participants, with MiRNA then being isolated from 200 μL of plasma using a miRNeasy Serum/Plasma Advanced Kit (Cat. No. 217204, Qiagen), following the manufacturer’s protocol. Plasma microRNAs were eluted in 20 μL of nuclease-free water. The concentrations of the extracted microRNAs were quantified using a ThermoFisher’s Qubit^®^ miRNA Assay Kit (Q32880, ThermoFisher Scientific Inc., Waltham, MA, USA). To synthesize the cDNA, 2 ng of total microRNAs taken from the samples were used in 20 μL reverse transcription reactions. The reverse transcription step was performed as follows: The Poly-A tail was added to the microRNA population using Poly-A polymerase, followed by cDNA synthesis with QuarkBio’s microRNA Reverse Transcription Kit (Quark Biosciences, Inc., Quark Biotechnology, Inc., Zhubei, Taiwan). qPCR was then performed utilizing the NextAmp™ Analysis System and mirSCAN™ V2 PanelChip^®^. For qPCR analysis, 0.15 ng cDNA was added to the QuarkBio qPCR master mix (Quark Biosciences, Inc.), and Q Station™ (Quark Biosciences, Inc.) was run according to the following cycling program: 95 °C for 36 s and 60 °C for 72 s over a total of 40 cycles.

### 4.3. Statistical Analysis

We analyzed the numbers for cases in both the control and graft dysfunction groups with regards to gender differences, age, age when receiving transplant, transplant duration, renal function, humoral immunity and cellular immunity. MiRNA expression data was normalized using quantile normalization, without internal control genes required. For further data preprocessing, miRNAs without amplification signals across all profiles were removed; the missing miRNA values for individual profiles were replaced with the maximum Delta Cq of all profiles. Based on the experimental design, the number of differentially expressed miRNA for each comparison will be identified. Clustering analysis was used to investigate differently expressed microRNAs between the control and graft dysfunction groups, based upon the threshold of fold change and *p*-value. Standard selection criteria used to identify differently expressed miRNAs were delta Cq ≥ |0.585| and an adjusted *p* < 0.05. These data were classified by gender for use in further studies. Data are shown as the mean for continuous variables and expressed as percentages or numbers for categorical data. Statistical significance was defined as *p* < 0.05.

## 5. Conclusions

Gender differences do exist in kidney graft dysfunction, with male patients experiencing better preservation of graft functions. Though we did not find any significant differences in complete blood counts and other immune factors between genders, differently expressed miRNAs have been noted, including miR-128-3p, miR-150-5p, miR-223-3p, and miR-26a. Overall, those miRNAs play essential roles in either enhancing or suppressing host immune responses. miRNAs regulate inflammatory pathways and can be important factors that lead to worse long term kidney graft function in females when compared to males. They can also act as potential predictive markers for graft survival. More studies should be performed in order to better understand the functions of each miRNA.

## Figures and Tables

**Figure 1 ijms-23-12832-f001:**
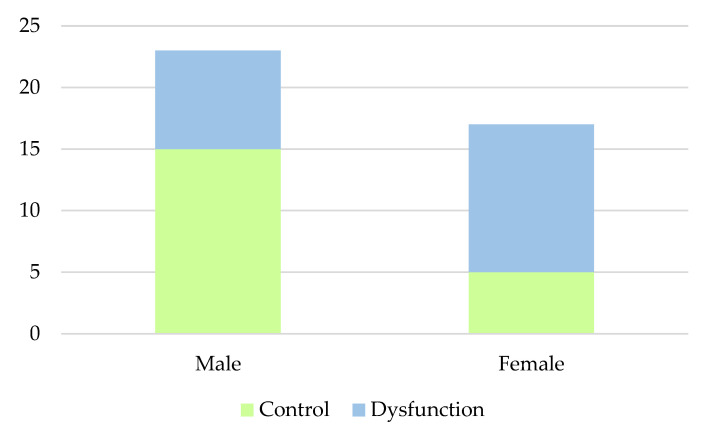
Distribution of Graft Dysfunction in Different Genders. Amongst all 40 patients, 12 out of 17 females and eight out of 23 males had previously undergone graft dysfunction. The graft dysfunction rate of the female KTx patients (71%) was shown to be significantly higher than that of male patients (35%), with a *p*-value < 0.05.

**Figure 2 ijms-23-12832-f002:**
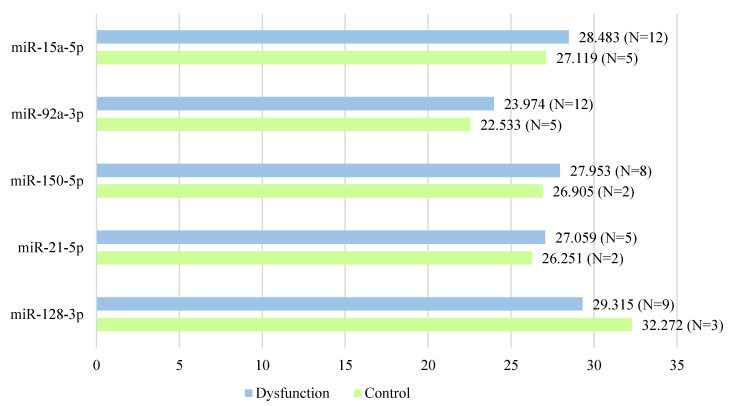
Mean Ct value of differently expressed miRNAs in female patients. In the female dysfunction group, miR-128-3p was up-regulated, while others were down-regulated when compared to the female control group.

**Figure 3 ijms-23-12832-f003:**
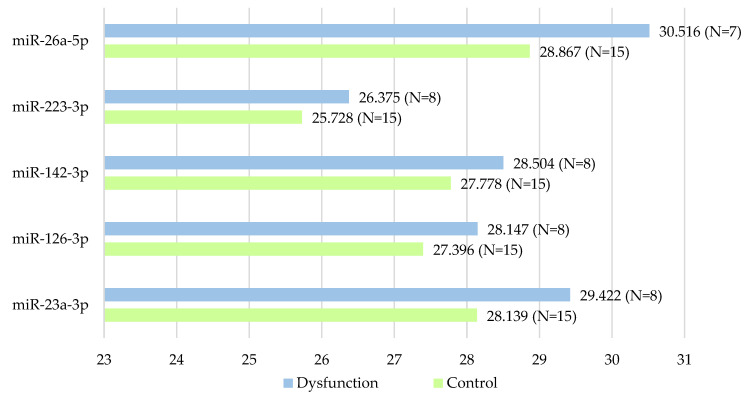
Mean Ct Value of differently expressed miRNAs in male patients. All of them were down-regulated in the male dysfunction group when compared to the male control group.

**Table 1 ijms-23-12832-t001:** Patient Characteristics of Different Genders. Demographic analysis of the 40 patients with KTx over a period of more than ten years is shown.

	Female		Male		*p*-Value
	Control	Dysfunction	Control	Dysfunction	
	*n* = 5	*n* = 12	*n* = 15	*n* = 8	
Age (years)	58.3 ± 5.4	55.5 ± 10.4	54.3 ± 9.8	49.1 ± 11.5	0.42
Age of KTx (years)	39.1 ± 5.5	37.1 ± 11.4	37.4 ± 12.3	34.3 ± 13.4	0.90
Vintage of KTx (years)	19.2 ± 6.1	18.4 ± 3.9	16.9 ± 4.0	14.8 ± 2.8	0.15
BUN (mg/dL)	13.6 ± 2.7	25.9 ± 10.1	16.3 ± 5.7	20.6 ± 3.3	0.002
Cr (mg/dL)	0.9 ± 0.1	1.5 ± 0.3	1.1 ± 0.1	1.7 ± 0.3	<0.001
eGFR (ml/min)	72.0 ± 9.0	40.8 ± 9.5	75.2 ± 7.2	48.6 ± 7.2	<0.001
UPCR (mg/mg)	101.2 ± 48.5	931.7 ± 2171.1	135.8 ± 83.9	316.9 ± 285.1	0.10
Plat (×10^3^/μL)	213.4 ± 27.6	246.9 ± 59.1	221.5 ± 63.4	229.9 ± 55.9	0.73
Hb (gm/dL)	13.8 ± 0.8	11.8 ± 1.5	14.1 ± 1.8	13.8 ± 1.4	0.01
WBC (/μL)	7162.0 ± 1900.2	7517.5 ± 2237.8	7012.7 ± 907.7	7433.8 ± 1436.8	0.78
NK (%)	18(17–19)	16 (9–23)	19 (1–28)	12 (5–19)	0.32
N/L ratio	2.0 (1.1–2.9)	2.9 (1.0–5.0)	2.6 (0.9–4.4)	3.7 (0.3–7.1)	0.60
P/L ratio (×1000)	6.9 (5.4–8.4)	10.7 (3.5–14.3)	8.7 (5.3–12.2)	11.0 (5.5–16.5)	0.80
T cell (%)	70 (58–82)	73 (67–79)	70 (61–79)	77 (65–89)	0.49
Activated T (%)	5 (3–7)	10 (4–16)	8 (5–11)	13 (1–25)	0.34
LEU-3A (%)	43 (38–48)	38 (28–48)	39 (29–49)	41 (35–47)	0.91
LEU-3A (μL^−1^)	969.8 ± 362.7	692.2 ± 233.3	762.4 ± 425.7	719.6 ± 366.4	0.60
LEU-2B (%)	27 (18–36)	33 (23–43)	30 (19–41)	34 (23–45)	0.45
LEU-2B (μL^−1^)	504.3 ± 335.9	650.6 ± 256.3	574.0 ± 291.4	615.3 ± 417.5	0.91
B-cell (%)	11 (5–17)	9 (3–15)	9 (3–15)	9 (1–17)	0.84
TH/TS (CD4/CD8)	1.7 (1.2–2.3)	1.4 (0.5–2.2)	1.5 (0.9–2.1)	1.4 (0.7–2.0)	0.45
C3 (mg/dL)	109.5 ± 15.0	111.5 ± 21.6	113.8 ± 23.0	122.2 ± 20.0	0.41
C4 (mg/dL)	28.8 ± 8.6	28.8 ± 5.8	29.5 ± 11.2	31.2 ± 13.6	0.98
CH50 (CAE)	101.33 ± 28.59	126.5 ± 3.54	108.88 ± 20.91	119.2 ± 22.11	0.66
IgG (mg/dL)	1113.6 ± 157.1	1228.7 ± 298.2	1135.3 ± 246.6	1119.8 ± 191.7	0.91
IgA (mg/dL)	320.5 ± 96.3	343.8 ± 230.2	249.8 ± 106.4	216.3 ± 92.7	0.32
IgM (mg/dL)	97.5 ± 30.4	136.1 ± 61.6	89.2 ± 38.7	98.6 ± 56.6	0.28
FK-506 (ng/mL)	5.1 ± 0.9	5.7 ± 0.9	5.1 ± 1.2	5.7 ± 1.3	0.57
Adagraf/prograf (mg/day)	3 ± 1.5	6.4 ± 2.4	3.7 ± 2.0	5.1 ± 3.6	0.03

Note: Control group: eGFR ≥ 60 mL/min; Graft dysfunction group: eGFR ≤ 60 mL/min. CD3 as the marker of T cell, HLA-DR as the marker of activated T cell, CD4 as the marker of LEU-3A, CD8 as the marker of LEU-2B, CD19 as the marker of B cell. All statistical analyses in this study were performed by R software (version 3.6). A Kruskal-Wallis Rank Sum Test from R built-in function (kruskal.test) was used to determine if there are statistically significant differences between independent groups.

**Table 2 ijms-23-12832-t002:** Differently expressed miRNAs in female patients. There were five miRNAs expressing with significant difference between the female control group and female dysfunction group: miR-128-3p, miR-21-5p, miR-150-5p, miR-92a-3p and miR-15a-5p.

	Numbers		Mean Ct		log2ratio	*p*-Value
microRNA ID	Control *n* = 5	Dysfunction *n* = 12	Control	Dysfunction		
miR-128-3p	2	8	32.272	29.315	2.957	0.0063
miR-21-5p	5	12	26.251	27.059	−0.808	0.0113
miR-150-5p	5	12	26.905	27.953	−1.048	0.0133
miR-92a-3p	5	12	22.533	23.974	−1.441	0.0172
miR-15a-5p	5	12	27.119	28.483	−1.364	0.0278

**Table 3 ijms-23-12832-t003:** Differently expressed miRNAs in male patients. There were five expressing with significant differences between the male control group and male dysfunction group: miR-23a-3p, miR-126-3p, miR-142-3p, miR-223-3p and miR-26a-5p.

	Number		Mean Ct		log2ratio	*p*-Value
microRNA ID	Control *n* = 15	Dysfunction *n* = 8	Control	Dysfunction		
miR-23a-3p	15	8	28.139	29.422	−1.283	0.0222
miR-126-3p	15	8	27.396	28.147	−0.751	0.0321
miR-142-3p	15	8	27.778	28.504	−0.726	0.0357
miR-223-3p	15	8	25.728	26.375	−0.647	0.0388
miR-26a-5p	15	7	28.867	30.516	−1.649	0.0395

**Table 4 ijms-23-12832-t004:** Significantly Different Expressions of Inflammation-related miRNAs in Dysfunction Patients. MiR-128-3p promotes inflammation while others are anti-inflammatory.

	Expression in Dysfunction Groups		Functions
	Female	Male	
MiR-128-3p	Upregulation	-	Proinflammation: Induces IL-6 and IL-17 production
MiR-150-5p	Downregulation	-	Anti-inflammation: Inhibits CD8+ T cells and mTOR, stimulates Treg differentiation
MiR-223-3p	-	Downregulation	Anti-inflammation: Suppresses TRAF6, NF-κB and IGF1R/PI3K pathway, reduces IL-6 secretion
MiR-26a	-	Downregulation	Anti-inflammation: stimulates Tregs/CD4+ T cells ratio, reduces IL-6 secretion

## Data Availability

The data presented in this study are available on request from the corresponding authors. The data are not publicly available due to ethical restrictions.

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
