# Peer review of "Gender Differences in microRNA Expressions as Related to Long-Term Graft Function in Kidney Transplant Patients"

_ijms, 2022, doi:10.3390/ijms232112832_

Round 1

Reviewer 2 Report

It has been recognized the important role that miRNAs play in many kidney diseases (Zhou and Yang. Implications of microRNA in kidney metabolic disorders. ExRNA 2020;2:4) and in the acceptance or rejection of organ transplantation (Alfaro R et al. MicroRNA expression changes in kidney transplant: Diagnostic efficacy of miR-150-5p as potential rejection biomarker, pilot study. J Clin Med 2021;10:2748). Then Ko et al. by analyzing immune-related serum miRNAs, aim to explain why graft dysfunction occurs more frequently in female kidney transplant patients than in male kidney transplant patients, which is interesting because there are few studies that address this situation.

The analysis made indicated that humoral and cellular immunity did not show any differences among groups studied. However it was interesting the fact that female and male dysfunction groups differently expressed each 5 miRNAs (miR-128-3p, miR-21-5p, miR-150-5p, miR-92a-3p and miR-15a-5p) with respect to its control groups and that in females 3 miRNAs were down-regulated and 2 miRNAs were up-regulated meanwhile in the male dysfunction group all miRNAs were down-regulated (miR-23a-3p, miR-126-3p, miR-142-3p, miR-223-3p and miR-26a-5p). Then there do exist gender differences in the microRNA expression of graft dysfunction patients.

The observed miRNAs that were differentially expressed are related to inflammatory pathways and then the authors suggest that the miRNAs that regulate inflammatory pathways (miR-128-3p, miR-150-5p) can be the responsible factors that worse long term kidney graft function in females.

Comments:

It is not clear whether effectively there are a lack of studies about the sex-differences effects on kidney transplant patients since authors mention this in the lines 60 - 61 but later they indicate the following  “To explain the reason for generally worse long-term outcomes in female KTx patients” (lines 188 and 189).

Once this is clarified, the authors could mention if its results are in agreement with other studies or if its data are original.

Authors mentioned that in transplanted patients the oldest age in female group was 30.1 years and the youngest age in male group was 34.3 years, is this correct? I got confused since authors previously indicate that similarly female group has the oldest mean age.

Please specify the statistical test and software used for comparisons made in Table 1, and also indicate the groups that were compared.

The symbol used for pro-inflammation and anti-inflammation columns in Table 4 are not clear. What do they indicate?
